# A Sample-driven Selection Framework: Towards Graph Contrastive Networks with Reinforcement Learning

## ABSTRACT

Graph Contrastive Learning (GCL) applied in real-world scenarios aims to alleviate label scarcity by harnessing graph structures to disseminate labels from a limited set of labeled data to a broad spectrum of unlabeled data. Recent advancements in amalgamating neural network capabilities with graph structures have demonstrated promising progress. However, prevalent GCL methodologies often overlook the fundamental issue of semi-supervised learning (SSL), relying on uniform negative sample selection schemes such as random sampling, thus yielding suboptimal performance within contexts. To address this challenge, we present GraphSaSe, a tailored approach designed specifically for graph representation tasks. Our model consists of two pivotal components: a Graph Contrastive Learning Framework (GCLF) and a Selection Distribution Generator (SDG) propelled by reinforcement learning to derive selection probabilities. We introduce an innovative strategy whereby the divergence between positive graph representations is translated into a reward mechanism, dynamically guiding the selection of negative samples during training. This adaptive methodology aims to minimize the divergence between augmented positive pairs, thereby enriching graph representation learning crucial for applications. Comprehensive experimentation across diverse real-world datasets validates the effectiveness of our algorithm, positioning it favorably against contemporary state-of-the-art methodologies.[1]

## CCS CONCEPTS

• **Computer systems organization** → **Embedded systems**; *Redundancy*; Robotics; • **Networks** → Network reliability.

## KEYWORDS

Graph contrastive learning, Graph neural networks, Reinforcement learning

## 1 INTRODUCTION

Graph neural networks (GNNs) have emerged as the leading approach for graph representation learning, owing to their capacity to iteratively aggregate information from neighboring nodes and edges, thereby effectively capturing both the structural properties of graphs and the features associated with nodes and edges [22, 42]. GNNs have demonstrated remarkable success across a wide range of

[1]Source code is available at: https://anonymous.4open.science/r/RL-CDC5/.

*ACM MM, 2024, Melbourne, Australia*
© 2024 Copyright held by the owner/author(s). Publication rights licensed to ACM.
ACM ISBN 978-x-xxxx-xxxx-x/YY/MM
https://doi.org/10.1145/nnnnnnn.nnnnnnn

domains, including social networks [27], protein molecules [6], and transportation networks [19]. However, the practical application of GNNs in contexts often encounters challenges due to the scarcity of labeled data [3]. For instance, in industrial settings, such as manufacturing processes or supply chain management, obtaining labeled data for graph learning can be arduous and resource-intensive. Additionally, in fields like chemical engineering, where understanding molecular structures is crucial for product development and optimization, acquiring labeled data for graph learning may involve costly and time-consuming experiments. To address these challenges, researchers are increasingly exploring self-supervised or unsupervised learning approaches for graph representation learning in applications. By leveraging techniques that require limited or no labeled data, such as contrastive learning or generative models, these methods aim to enhance graph learning efficiency and scalability in contexts. This shift towards unsupervised learning paradigms offers promising avenues for advancing graph-based solutions, where labeled data may be scarce or costly to obtain.

Due to their demonstrated effectiveness, graph contrastive learning (GCL) has emerged as a leading self-supervised approach, aiming to extract discriminative representations without relying on human annotations. These methodologies typically achieve this by learning representations that remain invariant to data augmentations, achieved by maximizing the agreement between embedding vectors derived from various distortions of the same graph [39]. Central to contrastive learning are two essential components: the concept of similar (positive) pairs $(x, x^+)$ and dissimilar (negative) pairs $(x, x^-)$ of data points. Subsequently, the training objective guides the learned representation to map positive pairs closer together while pushing negative pairs farther apart [26]. However, the effectiveness of these methods heavily relies on the design of these positive and negative pairs, which cannot fully exploit true similarity information due to the lack of supervision.

While existing literature has predominantly focused on refining positive pairs through diverse graph augmentation strategies like node dropping and edge perturbation, the significance of adept negative sample selection in graph contrastive learning (GCL) has often been overlooked. Surprisingly, in practical scenarios, superior or comparable test performance can be achieved by meticulously selecting relevant negative samples for GCL. Recent findings underscore the unequal contributions of individual samples to training, with lower-quality samples hindering progress and undermining overall performance. Consequently, the careful curation of high-quality negative samples emerges as pivotal for optimizing performance and safeguarding against training stagnation. In industrial contexts, the meticulous selection of negative samples holds significant promise for enhancing various processes. For instance, in manufacturing, precise negative sample selection can aid in optimizing production workflows and identifying potential inefficiencies. Similarly, in supply chain management, leveraging graph

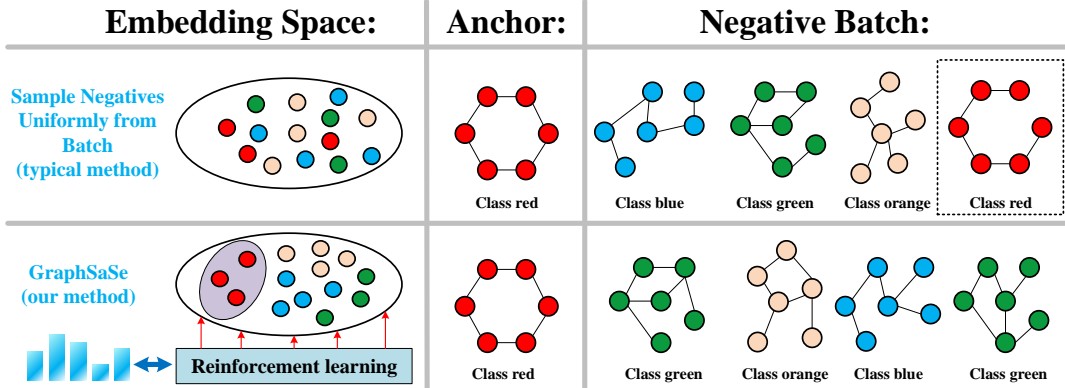

**Figure 1: Schematic illustration of negative sampling methods. Top row: randomly sampling negative examples from a batch apart from the anchor, but may even sample examples from the same class. Bottom row: utilizing reinforcement learning to adaptively select the negative samples that share a different label with the anchor.**

contrastive learning with adept negative sample selection can offer insights into network relationships among suppliers, manufacturers, and distributors, thereby enhancing decision-making processes and supply chain efficiency. Moreover, in fields like chemical engineering and materials science, where molecular structure analysis is paramount, effective negative sample selection can facilitate the prediction of material properties and the design of novel materials with tailored characteristics. Thus, refining negative sample selection methodologies holds potential for driving efficiency and innovation across diverse industrial domains. However, devising effective negative samples presents its own set of challenges. Current methodologies frequently resort to random sampling from batches, a practice depicted in **Figure 1**. Yet, this approach risks the inadvertent inclusion of invalid-negative samples sharing labels with the anchor, potentially exacerbating the training process. Although some studies, like Cuco [3], recognize the importance of negative samples, their methods entail comparing entire datasets during training, demanding substantial computational resources. In light of these considerations, a pertinent inquiry arises: **is there a more refined approach to selecting appropriate negative samples for graph contrastive learning?**

We should eliminate the influence of "invalid" negative samples in graph contrastive learning. Careful treatments and additional strategies are usually needed to circumvent invalid-negative samples and pick hard negative ones [26]. Thus, *making graph contrastive learning self-adapted to select the negative samples* is a **challenge** and has become increasingly important to learn the representations of entire graphs. Considering that data selection is, in general, a combinatorial optimization problem with complexity, it is impossible to try all possible combinations of training instances [21]. In this way, we hope that the previous data selection of GCL should influence the later data decisions. In this case, reinforcement learning (RL) can be an appropriate vehicle, which has proved to be a prospective approach for data training selection in leveraging pertinent data [5]. Such reinforcement mechanisms make GCL especially appealing in selecting a significant portion of training

data for graphs. However, how to effectively take advantage of RL to benefit GCL remains largely under-explored, which is still a significant challenge.

In this paper, we present Graph Sample Selection Contrastive Learning (GraphSaSe), a novel approach that integrates reinforcement learning (RL) with graph contrastive learning (GCL) to address the aforementioned challenge effectively. By dynamically selecting appropriate negative samples from training data, GraphSaSe enhances graph representations, making it particularly suitable for real-world applications. Comprising a graph contrastive learning framework (GCLF) and a selection distribution generator (SDG), GraphSaSe optimizes the contrastive objective function by maximizing similarity between graph augmentations (positive views) while minimizing similarity to other samples within a batch (negative views). Through comprehensive experimentation, GraphSaSe demonstrates its efficacy in enhancing graph representation learning across a spectrum of applications. Its ability to dynamically adapt to diverse datasets and optimize representation learning makes it a promising tool for addressing challenges in real-world scenarios.

The main contributions of this work are summarized as follows:

- We propose an effective methodology termed Graph Sample Selection Contrastive Learning, denoted as GraphSaSe, designed to refine graph representation techniques to show its efficacy in enhancing graph representation learning across a spectrum of applications. To the best of our knowledge, we are the first to attempt to utilize RL to select the appropriate negative samples in graph contrastive learning.

- In our novel approach, we ingeniously translate the divergence observed between positive graph representations into a rewarding mechanism. Additionally, we meticulously devise a selection distribution generator, enabling adaptive selection of negative samples to effectively steer the training process.

- Extensive experimentation conclusively demonstrates the remarkable superiority of our approach over state-of-the-art baselines. Our method consistently achieves statistically significant

improvements across real-world datasets, affirming its efficacy and robustness.

## 2 RELATED WORK

### 2.1 Graph Convolutional Network.

Graph Convolutional Networks (GCNs) represent a groundbreaking extension of the powerful Convolutional Neural Networks (CNNs) into the realm of graph-structured data [18]. The applications of GCNs extend far beyond traditional domains into various industrial contexts. For example, in smart grid management, GCNs can model complex power distribution networks, optimizing energy flow and predicting potential failures. Hence, the integration of GCNs with contrastive learning not only advances the field of graph representation learning but also holds immense promise for enhancing industrial processes and innovation. This process seamlessly integrates connectivity patterns and feature attributes inherent in graph-structured data, leading to superior performance over many state-of-the-art methods on various benchmarks [37], including L2-GCN [40] and AM-GCN [33]. To address the scalability challenge posed by spectral-based GCNs when dealing with large-scale images, spatial-based GCNs have emerged rapidly [8]. For instance, PATCHY-SAN addresses this issue by sorting nodes and selecting a fixed number of adjacent points for graph convolution, inspired by GCN principles [24]. Similarly, the Graph Attention Network (GAT) introduces an attention mechanism to define graph convolutional operations [32]. In this study, we harness the potential of contrastive learning techniques to tap into the rich information contained within unlabeled graphs. By applying contrastive learning, we aim to extract meaningful representations from unlabeled data, thus enhancing the performance and scalability of graph-based models, which is crucial for various real-world applications.

### 2.2 Graph Contrastive Learning.

Existing graph contrastive learning methods [4, 14, 46] train an encoder to measure the divergence in latent space by contrasting samples from a distribution that contains depict statistical dependencies of interest and those that do not [9]. Their basic idea is to make multiple views from the same instance under various graph transformations agree with each other to optimize model parameters. Despite the similarity in principle, these methods are elaborately designed and differ from each other in various aspects, such as network architecture, augmented view design, and contrastive objectives. For instance, InfoGraph [28] maximizes the mutual information between the graph-level representation and the representations of substructures of different scales. GraphCL [39] first designs four types of graph augmentations to incorporate various priors and then systematically study the impact of various combinations of graph augmentations. The theoretical analysis sheds light on the reasons behind their success [46]. Objectives used in these methods can be seen as designing different graph augmentation strategies to enhance the graph representation, while little effort from previous works for negative sample selection strategy. In contrast, our method adopts the RL to optimize the data selection process, which can be trained with GNNs in an end-to-end manner.

### 2.3 Reinforcement Learning.

Reinforcement learning (RL) is one of the effective machine learning paradigms, aiming to learn how to make decisions for maximizing the cumulative future rewards [43]. Previous empirical studies [5, 21] have shown that RL generally does not require numerous labeled datasets but obtains samples for training through ongoing interactive trial and error among the environment, which is closer to the human learning process. Recent trends in RL field is to combine CNNs with RL algorithms for solving high-dimensional complex problems, such as object localization [45] and object detection [11]. Unlike their works, we leverage RL to guide GCL and narrow the divergence between the augmented positive pairs, so as to further improve graph representations.

## 3 PRELIMINARY

**Graph Neural Networks.** Graph neural networks (GNNs) have emerged as a promising approach for analyzing graph-structured data recently [39]. We use $\mathcal{G} = (\mathcal{V}, \mathcal{E})$ to denote an undirected graph and define $\mathbf{X} \in \mathbb{R}^{N \times D}$ as the feature matrix, where $\mathcal{V} = \{v_i\}_{i=1}^N$ represent a set of $N$ nodes, $\mathcal{E} = [e_{i,j}] \in \mathbb{R}^{N \times N}$ denotes the adjacency matrix, $x_i = \mathbf{X}[i,:]^T$ is the $D$-dimensional attribute vector of the node $v_i \in \mathcal{V}$. GNNs use a neighborhood aggregation approach, whose propagation process can be writen as:

$$\mathbf{h}_i^k = \Theta\left(\mathbf{h}_i^{k-1}, \delta_{j \in \mathcal{N}_i} \phi\left(\mathbf{h}_i^{k-1}, \mathbf{h}_j^{k-1}, e_{i,j}\right)\right) \tag{1}$$

where $\mathbf{h}_i^k$ represents the output of the $k$-th network layer with $\mathbf{h}_i^0 = x_i$ i.e, the initial feature of node. $\mathcal{N}_i$ is a set of vertices adjacent to $v_i$. $e_{i,j}$ represents the edge feature between node $i$ and $j$, which is an option. $\delta$ denotes differentiable, permutation invariant function, such as sum or mean. $\Theta$ and $\phi$ represent differentiable functions or network layers, such as multi-layer perceptron (MLP). After the $K$-layer propagation, the output embedding for $\mathcal{G}$ is summarized on layer embeddings through the READOUT function:

$$f(\mathcal{G}) = \mathbf{READOUT}\left(\left\{\mathbf{h}_i^k : v_i \in \mathcal{V}, k \in K\right\}\right) \tag{2}$$

**Problem Definition.** Prior to going further, we first provide the problem definition used in this paper. For self-supervised graph representation learning, a set of unlabeled graphs $\mathbb{G} = \{\mathcal{G}_1, \mathcal{G}_2, \cdots, \mathcal{G}_n\}$ are given, and we aim to learn a $d$-dimensional vector $z_{\mathcal{G}_i} \in \mathbb{R}^d$ for each graph $\mathcal{G}_i \in \mathbb{G}$ under the guidance of the data itself, whose representation can be used in the downstream tasks.

## 4 METHODOLOGY

The main goal of this paper is to train a GNN encoder with an RL-based adaptive sampling strategy. As shown in **Figure 2**, our method consists of two critical components: A graph contrastive learning framework (GCLF), and selection distribution generator (SDG) in the following.

### 4.1 Graph Contrastive Learning Framework

This section discusses how to perform contrastive learning on graphs. We aim to capture the intrinsic patterns and properties of input graph data without using human-provided labels. The process of GCLF is generating augmented examples from original

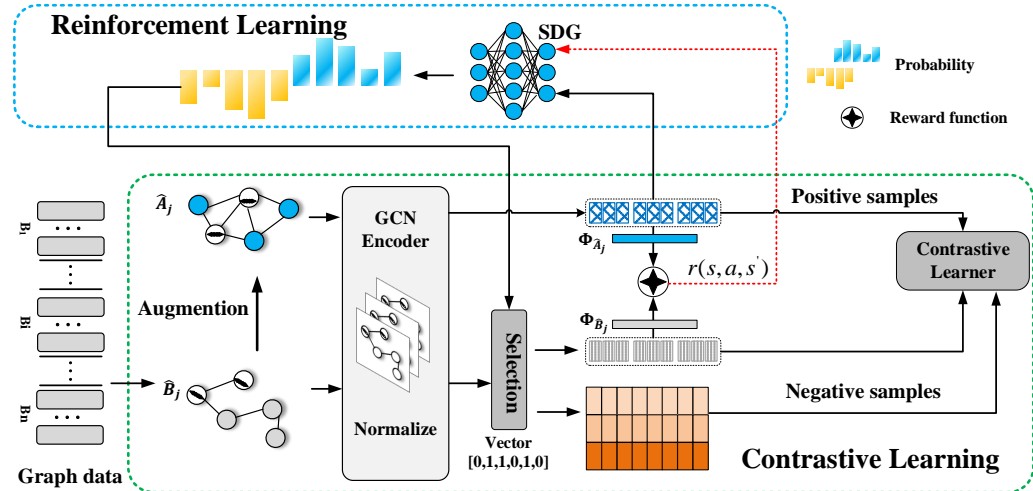

**Figure 2: The architecture of our GraphSaSe approach, with a graph contrastive learning framework and a selection distribution generator. All black solid arrows refer to data flow, while the red dashed arrow denotes reward.**

graph data, learning their representations with a graph encoder, and selecting negative samples for the final graph contrastive learning.

**Graph Augmentations.** Previous work [39] suggests that structural feature expansion can effectively improve the performance of graph classification. Data augmentation aims to create novel and realistically rational data by applying certain transformations without affecting the semantics label. To perform GraphSaSe on graphs, given a original graph $\mathcal{G} \in \{\mathcal{G}_q : q \in Q\}$ in the dataset of $Q$ graphs, we perform graph augmentation to formulate augmented graphs $\hat{\mathcal{G}}$ from the original graphs satisfying: $\hat{\mathcal{G}} \sim \mathcal{M}(\hat{\mathcal{G}}|\mathcal{G})$, where $\mathcal{M}(\cdot|\mathcal{G})$ is the augmentation distribution conditioned on the original graph, which is pre-defined, representing the human prior for data distribution. We focus on graph-level augmentations and adopt four basic data augmentation strategies to construct positive pairs of graphs [39], which are *node dropping, edge perturbation, attribute masking* and *subgraph*, respectively. More details can be seen in **Appendix A**. After that, augmented graphs $\hat{\mathcal{G}}$ could preserve task-relevant information, while simultaneously minimizing irrelevant information across views.

**Graph Encoder.** The purpose of training a graph encoder is to extract the most important information that, is data self-correlation, preserve task-relevant properties, and prevent the model from learning results that may lead to brittle representations. Notice that our method allows various choices of network architecture without any constraints. Formally, considering the general GNN framework in **Table 1**, we take GIN [37] as an example to instantiate the GNN encoder in the anomaly problem. GIN calculates the representation for each node via a sum-like neighborhood aggregation function. We initialize $h_i^{(0)} = x_{v_i}$. After $k$ rounds of aggregation, each node $v_i \in \mathcal{V}$ obtains its representation $h_i^{(k)} \in \mathbb{R}^d$, aggregated from their neighbors $\mathcal{N}_{v_i}$. The other three GNN frameworks have similar neighborhood aggregation and can be generalized to more datasets for graph embeddings. Finally, we obtain the output embedding

$f(\mathbf{H})$ through the READOUT function (See Eq.(2)). Then a multi-layer perceptron (MLP) is adopted, e.g., $z_i = \mathbf{MLP}(f(\mathbf{H}))$, for the graph-level downstream task (e.g., classification or regression).

**Table 1: Neighborhood aggregation schemes.**

| Methods | ‖ | Aggregation and combination functions for round $k(1 \leq k \leq K)$ |
|---|---|---|
| General GNN | ‖ | $h_i^{(k)} = \mathbf{COMBINE}^{(k)}\left(\left\{h_i^{(k-1)}, \mathbf{AGGREGATE}^{(k)}(\{h_j^{(k-1)} : v_j \in \mathcal{N}_{v_i}\})\right\}\right)$ |
| GCN [16] | ‖ | $h_i^{(k)} = \Theta\left(\sum_{v_j \in \mathcal{N}_{v_i} \cup \{v_i\}} \frac{1}{\sqrt{(\|\mathcal{N}_{v_i}\|+1)\cdot(\|\mathcal{N}_{v_j}\|+1)}} \cdot \mathbf{W}^{(k-1)} \cdot h_{v_j}^{(k-1)}\right)$ |
| GIN [37] | ‖ | $h_i^{(k)} = \Theta\left((1+\epsilon)\cdot h_i^{(k-1)} + \sum_{v_j \in \mathcal{N}_{v_i}} h_j^{(k-1)}\right)$ |
| GraphSAGE [8] | ‖ | $h_i^{(k)} = \Theta\left(\mathbf{W}^{(k-1)} \cdot \left[h_i^{(k-1)} \| \sum_{v_j \in \mathcal{N}_{v_i}} h_j^{(k-1)}\right]\right)$ |
| GAT [32] | ‖ | $h_i^{(k)} = \Theta\left(\sum_{v' \in \mathcal{N}_v \cup \{v\}} a_{i,j}^{(k-1)} \cdot \mathbf{W}^{(k-1)} \cdot h_j^{(k-1)}\right)$ |

## 4.2 Selection Distribution Generator

Recent GCL works fail to consider the class information, leading to more miniature discriminative graph representations issue [44]. More concretely, without access to labels, the randomly selected negative samples may have the same label as the positive sample in a situation, causing a performance drop. Thus, we attempt to exploit these probabilities to opt for the negative samples to widen the differences between their classes. Specifically, a sequence of 0-1 states is obtained through Bernoulli distribution to sample negative instances. However, how to produce the selection probabilities matters, which will discuss in the following.

Obviously, a multi-layer perceptron (MLP) model is used as the SDG, which learns the selection policy optimized by the reward from the divergence between positive graphs representation based on RL. The reward can be trained by SDG and then generate selection probabilities that stay away from positive samples. In what follows, we will introduce a novel negative sampling method by adopting RL. The main idea is to opt for negative samples far away from positive samples during training based on the

selection probabilities. For a more rigorous description, given a graph dataset $\mathcal{D} \in \{x_1, x_2, \cdots, x_n\}$, we divide it into $N$ mutually exclusive batch sets marked as $\mathcal{D} = \{B_1, B_2, \ldots, B_N\}$, where $B_j = \left\{x_{(j-1)n/N+1}, x_{(j-1)n/N+2}, \ldots, x_{jn/N}\right\}$, $j \in \{1, 2, \ldots, N\}$. $n$ is the total number of a dataset and we marked $|B_j|$ as the number of samples in one batch set. Subsequently, we denote the collection of graph representations with the aforementioned graph encoder for the batch sets as $\Phi_{B_j} = \left\{b_1^j, b_2^j, \cdots, b_{|B_j|}^j\right\}$, where $b_l^j$ ($l = 1, 2, \cdots, |B_j|$) is the vector of the $l$-th sample in $B_j$. Then SDG maps the graph representation $\Phi_{B_j}$ or $\Phi_{A_j}$ ($A_j$ is the augmented graphs of $B_j$) to generate a selection vector $\Gamma_{B_j} = \left\{v_1^j, v_2^j, \cdots, v_{|B_j|}^j\right\} \in \mathbb{R}^{1 \times |B_j|}$, which represents the probability for each instance on the confidence of select. Generally speaking, RL is defined by specifying three ingredients: *state, action*, and *reward*, described as follows.

**State.** We then define a state collection marked as $state = \{s_1, s_2, \ldots, s_j, \ldots, s_N\}$, which includes a collection of states for all $j$ with respect to $N$ batch sets, where each $s_j$ indicates a state including selected positive graph instances $\hat{B}_j$ sampled from $B_j$ according to the distribution vector $\Gamma_{B_j}$, For simplicity we use $\Phi_{\hat{B}_j}$ and $\Phi_{\hat{A}_j}$ ($\hat{A}_j$ is the augmented graphs of $\hat{B}_j$) to represent state $s_j$.

**Action.** For each state, the action space $\mathbf{A}$ is a 0-1 judgment of how to select positive and negative samples. We take instances (1) to choose positive pairs, while instances (0) for negative ones. We get a list of actions $a = \{a_l\}_{l=1}^{|B_j|} \in \{0, 1\}^{|B_j|}$ from $\Gamma_{\hat{B}_j}$, whose process of assigning the value, i.e., 1 or 0, to $l$-th element of action $a$ can be formulated by sampling from a Bernoulli distribution. After each action, the framework gives new $\Phi_{\hat{B}_j}$, then the satate $s_j$ is changed to $s_j'$. The policy is defined as $P_\omega (a \mid s)$.

**Reward.** Reward mechanism is the core of RL, here we set a reward $r (s, a, s')$ to assess the distance between $\Phi_{\hat{B}_j}$ and $\Phi_{\hat{A}_j}$ in the current state ($s'$) and its previous state ($s$):

$$r(s, a, s') = d(\Phi_{\hat{B}_{j-1}}^s, \Phi_{\hat{A}_{j-1}}^s) - \gamma d(\Phi_{\hat{B}_j}^{s'}, \Phi_{\hat{A}_j}^{s'}) \tag{3}$$

where $d(\cdot, \cdot)$ is a distribution discrepancy measurement (**DDM**) implemented by different information-bearing functions, discussed in the following part. $\gamma \in (0, 1)$ is a discounting constant that decreases the impact from future distribution divergences. Note that our reward is related not only to the current status of the $j$-th selected batch set $\hat{B}_j$, but also to the status of the $(j-1)$-th selected samples $\hat{B}_{j-1}$. Eq.(6) is conducted in a sequential manner based on two adjacent batch sets $B_{j-1}$ and $B_j$, of which $\Phi_{\hat{B}_j}^{s'}$ is impacted by $\Phi_{\hat{B}_{j-1}}^s$ via parameters of the GCN Encoder updated by $\hat{B}_{j-1}$. When better instances are selected, the reward is then expected to produce a higher value because the measurement for the previous state $d(\Phi_{\hat{B}_{j-1}}^s, \Phi_{\hat{A}_{j-1}}^s)$ is supposed to give a larger distance between $\Phi_{\hat{B}_{j-1}}^s$ and $\Phi_{\hat{A}_{j-1}}^s$ than that for the current state. Our design aims to ensure that the actions obtained from SDG can also have a particular gain on the selection of the next batch set, rather than only affecting the current batch set, which makes the selection stable.

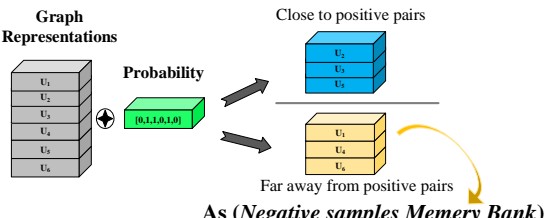

**Figure 3: Selecting samples, where $u$ represents a graph.**

**Distribution Discrepancy Measurements.** To measure each selected original graphs $\hat{B}_j$ and augmented graphs $\hat{A}_j$, let $P = (p_1, \cdots, p_n)$ be the normalized element-wise average of $\Phi_{\hat{B}_j}$ and $Q$ the average of $\Phi_{\hat{A}_j}$ similarly. We calculate $d(\cdot, \cdot)$ with the following three different distribution discrepancy measurements (DDM) for reward function, namely MMD, RÉNYI, and JS.

**MMD**: The maximum mean discrepancy [1],

$$d(P, Q) = \|P - Q\| \tag{4}$$

**RÉNYI**: The symmetric Rényi divergence [25], we set $\alpha = 0.99$ following [30].

$$d(P, Q) = \frac{1}{2}\left[Ry(P, \frac{P+Q}{2}) + Ry(Q, \frac{P+Q}{2})\right] \tag{5}$$

$$Ry(P, Q) = \frac{1}{\alpha - 1} \log\left(\sum_{i=1}^{n} \frac{p_i^\alpha}{q_i^{\alpha-1}}\right) \tag{6}$$

**JS**: The Jensen-Shannon divergence [20],

$$d(P, Q) = \frac{1}{2}\left[D_{KL}(P\|\frac{P+Q}{2}) + D_{KL}(Q\|\frac{P+Q}{2})\right] \tag{7}$$

$$D_{KL}(P\|Q) = \sum_{i=1}^{n} p_i \log \frac{p_i}{q_i} \tag{8}$$

**Note that** our method can adopt any related graph difference measurements, not limited to the MMD, RENYI, and JS. Just examples in this part.

**How to select negative samples adaptively?** As shown in **Figure 3**, for instance, suppose a graph batch is 6, and a 0-1 selection vector [0, 1, 1, 0, 1, 0] is obtained through SDG based on RL. In this way, a batch of 6 graphs will be divided into 3 graphs closed to positive pairs and 3 graphs far away from positive pairs as negative samples memory bank for the following contrastive learning process. Note that due to the award generated by the divergence between the positive pairs, such that samples with the same label will roughly gather in feature space finally and we can use this information to help samples in the same class closer. Finally, choosing the graph representation far away from the positive pairs as corresponding negative samples memory bank.

## 4.3 Model Optimization

The following object is optimized to obtain the optimal distribution generation policy:

$$J(\omega) = \mathbb{E}_{p_\omega(a|s)} \left[ \sum_{j=1}^{N} \gamma^{j-1} r\left(s_j, a_j\right) \right] \tag{9}$$

Then the parameters of the SDG, i.e., $\omega$, is updated via policy gradient [17] by:

$$\omega \leftarrow \omega + \tau \nabla_\omega \tilde{J}(\omega) \tag{10}$$

where $\tau$ is the discounting learning rate and can be self-adapted by optimizer, such as Adam. The gradient $\nabla_\omega J(\omega)$ is approximated by:

$$\nabla_\omega \tilde{J}(\omega) =$$
$$\frac{1}{T} \sum_{k=1}^{T} \sum_{j=1}^{N} \nabla_\omega \log \pi\omega \left(a_j^k \mid s_j^k\right) \sum_{j=1}^{N} \gamma^{j-1} r\left(s_j^k, a_j^k\right) \tag{11}$$

with $j$ referring to the $j$-th step (corresponding to the $j$-th batch set) in Reinforcement Learning, and $k$ is the $k$-th selection process to estimate $\nabla_\omega J(\omega)$, which is updated after every $T$ times of selection over all $N$ data batch sets.

To maximize the consistency between positive pairs $\{z_i, z_j\}$ compared with negative pairs $\{z_k\}_{k=1}^{K}$, we adopt the noise-contrastive estimation loss [31]:

$$\mathcal{L}_{NCE} =$$
$$- \log \frac{exp(sim(z_i, z_j)/\tau)}{\sum_{k=1}^{K} exp(sim(z_i, z_j)/\tau) + exp(sim(z_i, z_k)/\tau)} \tag{12}$$

where $\tau$ denotes the temperature parameter. $K$ denotes the number of negative sample in one batch. To simplify the calculation, we use dot product as the similarity metric function $sim(\cdot, \cdot)$. Finally, we combine $\mathcal{L}_{NCE}$ and $J(\omega)$ to get the total loss function:

$$\mathcal{L}_{total} = \mathcal{L}_{NCE} + \lambda J(\omega) \tag{13}$$

where $\lambda$ is a hyper-parameter to control the magnitude of reinforcement learning task. The entire learning process is described in **Appendix B**.

## 5 EXPERIMENTS

In this section, we aim to showcase the rapid and robust learning capabilities of our self-supervised model, GraphSaSe, in the realm of graph representation, which holds immense significance for various industrial applications. To achieve this, we conduct extensive comparisons with state-of-the-art methods (SOTAs) across unsupervised and transfer learning scenarios for graph classification tasks, providing insights into GraphSaSe's effectiveness in industrial contexts. Through these comparisons, we seek to highlight the superior performance and efficiency of GraphSaSe in learning high-quality representations from graph-structured data, thus bolstering its potential impact on industrial processes and innovation.

### 5.1 Unsupervised Representation Learning

**Datasets.** To evaluate our model, we conduct experiments on eight real-world datasets [2] in three fields, including three molecules

---

[2]Eight widely used datasets are publicly available at https://ls11-www.cs.tudortmund.de/staff/morris/graphkerneldatasets.

datasets: MUTAG, PTC, NCI-1, three social network datasets: REDDIT-BIN, IMDB-BINARY (IMDB-B), IMDB-MULTI (IMDB-M), and two bioinformatics datasets: PROTEINS, D&D. More details can be found in **Appendix C**.

**Baseline Methods.** The following models, which are the advanced and closely related works, including **HGCL** [15], **AD-GCL** [29], **SimGRACE** [34], **GraphMAE** [12], **LaGraph** [35], **CuCo** [3], **GraphCL** [39], **InfoGraph** [28], **JOAO** [38], **LG2AR** [10], are used as representative baselines to evaluate the performance of the proposed model. More details can be found in **Appendix D**.

**Experimental Setting.** For graph classification tasks, we adopt the same procedure of previous works [3, 23, 28, 39] to make a fair comparison and used 10-fold cross-validation accuracy to report the classification performance. Experiments are repeated 5 times. For some classical supervised learning algorithms, we report results from previous papers since we have the same experimental setup. If results are not previously reported, we implement them and conduct a hyper-parameter search according to the original paper. For all methods, the parameters of downstream classifiers are independently tuned using cross-validation on training folds of data. The best average classification accuracy is reported for each method. The classification accuracies are computed using **LIBSVM** [2], and the $C$ parameter was selected from $\left\{10^{-3}, 10^{-2}, \cdots, 10^{2}, 10^{3}\right\}$. For our proposed model, we adopt a four-layer GNNs with 32-dimensional hidden units and a sum pooling readout function for performance comparisons. To measure each original and augmented graph, we select RÉNYI to calculate the DDM for a reward.

#### 5.1.1 Overall Comparison.
**Table 2** shows the results of evaluating unsupervised graph-level representations using downstream graph classification tasks on multiple fields datasets. We have the following observations: (1) Overall, our proposed GraphSaSe consistently achieves top-tier performance across all datasets and architectures, affirming the effectiveness of our approach. Notably, on the MUTAG dataset, where existing baselines have already achieved high performance, GraphSaSe continues to push the boundaries further. This underscores the robustness and efficacy of our method in selecting negative samples for graph contrastive learning, even in scenarios where performance levels are already elevated. (2) Additionally, we observe that our method consistently outperforms recent unsupervised baselines. This superiority can be attributed to our deliberate consideration of the advantages stemming from high-quality and well-selected negative samples, which play a pivotal role in the training process. While approaches like CuCo also address this limitation by selecting negative samples, their reliance on comparisons among all graphs in the dataset demands substantial computational resources. As a result, our approach offers a more efficient and resource-friendly alternative while achieving competitive performance.

#### 5.1.2 Analysis on Reward.
Our reward is based on RL, and its formula is :

$$r\left(s, a, s'\right) = d(\Phi_{\hat{B}_{j-1}}^{s}, \Phi_{\hat{A}_{j-1}}^{s}) - \gamma d(\Phi_{\hat{B}_j}^{s'}, \Phi_{\hat{A}_j}^{s'}) \tag{14}$$

where we assess the distance between the current state (e.g., $d(\Phi_{\hat{B}_j}^{s'}, \Phi_{\hat{A}_j}^{s'})$) and its previous state (e.g., $d(\Phi_{\hat{B}_{j-1}}^{s}, \Phi_{\hat{A}_{j-1}}^{s})$). We use the $\gamma$ to control the magnitude of the current status for getting better reward

Table 2: Graph classification accuracy (%) of our method compared with state-of-the-art graph classification methods on three fields datasets. Their results are obtained from the corresponding original papers. The best performers are shown in bold.

| Methods | Molecules | | | Social networks | | | Bioinformatics | |
|---|---|---|---|---|---|---|---|---|
| | MUTAG | PTC | NCI-1 | REDDIT-BIN | IMDB-B | IMDB-M | PROTEINS | D&D |
| HGCL | 90.1 ± 0.8 | 59.3 ± 2.1 | 77.3 ± 1.2 | 83.4 ± 1.6 | 73.9 ± 0.7 | 51.3 ± 0.5 | 75.5 ± 0.5 | 79.2 ± 0.6 |
| InfoGraph | 89.0 ± 1.1 | 61.6 ± 1.4 | 76.2 ± 1.0 | 82.5 ± 1.4 | 73.0 ± 0.8 | 49.6 ± 0.5 | 74.4 ± 0.3 | 72.8 ± 1.7 |
| GraphCL | 86.8 ± 1.3 | 59.2 ± 2.5 | 77.8 ± 0.4 | 87.5 ± 0.8 | 71.1 ± 0.4 | 48.6 ± 1.2 | 74.3 ± 0.4 | 78.6 ± 0.4 |
| AD-GCL | 89.7 ± 1.1 | 57.2 ± 1.3 | 75.1 ± 0.3 | 85.8 ± 0.9 | 72.3 ± 0.6 | 49.8 ± 0.6 | 73.9 ± 0.4 | 77.3 ± 0.8 |
| SimGRACE | 89.0 ± 1.3 | 59.1 ± 0.7 | 79.1 ± 0.4 | 86.2 ± 1.1 | 71.3 ± 0.7 | 48.2 ± 0.9 | 75.3 ± 0.1 | 76.2 ± 1.3 |
| JOAO | 87.7 ± 0.8 | 55.3 ± 1.4 | 74.8 ± 0.7 | 86.4 ± 1.5 | 70.8 ± 0.3 | 49.3 ± 0.8 | 74.6 ± 0.4 | 77.4 ± 1.2 |
| LG2AR | 90.0 ± 0.6 | 56.8 ± 1.6 | 75.6 ± 0.9 | 91.8 ± 0.4 | 74.5 ± 0.6 | 51.9 ± 0.3 | 75.0 ± 0.5 | 79.1 ± 0.3 |
| GraphMAE | 88.1 ± 1.2 | 60.2 ± 2.3 | 80.4 ± 0.3 | 89.2 ± 0.9 | 75.5 ± 0.6 | 51.6 ± 0.5 | 75.3 ± 0.3 | 78.1 ± 1.3 |
| LaGraph | 90.2 ± 1.1 | 58.1 ± 1.9 | 79.9 ± 0.5 | 90.4 ± 0.3 | 73.7 ± 0.9 | 49.2 ± 1.1 | 75.2 ± 0.4 | 77.2 ± 1.5 |
| CuCo | 90.5 ± 0.9 | 58.9 ± 1.8 | 79.2 ± 0.5 | 88.6 ± 0.5 | 71.6 ± 2.2 | 48.7 ± 1.8 | 75.9 ± 0.5 | 79.2 ± 1.1 |
| **Ours** | **93.3 ± 1.2** | **64.4 ± 1.9** | **81.5 ± 1.4** | **92.5 ± 0.9** | **75.8 ± 1.3** | **53.9 ± 2.2** | **77.1 ± 0.8** | **82.3 ± 0.8** |

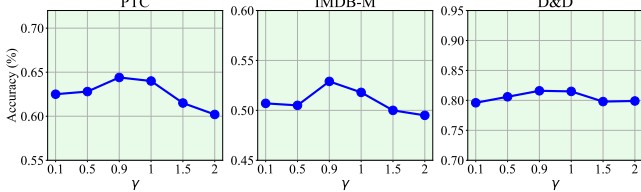

Figure 4: Analysis on the value $\gamma$ of reward on three datasets.

Table 3: Graph performance

| Models | MUTAG | NCI-1 | PROTEINS | Time (s)/Epoch |
|---|---|---|---|---|
| GCN | 85.6% | 76.2% | 75.1% | 0.29/3.26/1.32 |
| GraphCL | 86.8% | 77.8% | 74.3% | 0.49/8.23/3.15 |
| CuCo | 90.5% | 79.2% | 75.9% | 0.51/49.17/3.89 |
| **Ours** | **93.3%** | **81.5%** | **77.1%** | 0.38/7.61/2.88 |

$r(s, a, s')$. Thus, we design three simple reward functions by changing the value of $\gamma$. (1) **Reward A**: We set ($\gamma = 0.5, 0.1$) to conduct the situation that \*reward\* goes up. (2) **Reward B**: We set ($\gamma = 1.5, 2$) to conduct the situation that \*reward\* goes down. (3) **Reward C**: We set ($\gamma = 1$) to conduct the situation that the current status is equally important as the previous status.

As for our data selection, we set $\gamma = 0.9$ that the current status is slightly smaller than the previous status. We choose datasets: PTC, IMDB-M, D&D as examples and conduct experiments. As exhibited in **Figure 4**, when we set $\gamma = 0.9$, we get the best performance. The main reason is that $\gamma = 0.9$ puts the \*reward\* in a state of dynamic equilibrium, in which the \*reward\* goes down or up to select the balanced samples (e.g., the balance positive and negative samples). Although $\gamma = 1$ also has the same effect, $\gamma = 0.9$ has the trend that the reward is expected to produce a higher value which the previous state can affect the current state.

*5.1.3  **Efficiency Analysis**.* As shown in **Table 3**, we can see that our model yields excellent performance over the up-to-date baselines and has a 2-4% relative improvement on three datasets (e.g., we choose MUTAG, NCI-1, PROTEINS as examples). Furthermore,

we perform an additional experiment to evaluate the training efficiency of all models. The values of the last column in **Table 3** represent the training time of models in one epoch on three graph datasets. Unfortunately, the efficiency of our model is not outstanding, but it gets ahead of some baselines. The main reason is that our model employs reinforcement learning to select samples for better training. Lastly, we simply apply RL to generate selection probabilities, which require less computational cost and can achieve greater performance improvements, which indeed is also a bright spot of our method.

*5.1.4  **Ablation Study**.* Our GraphSaSe pairs reinforcement learning with contrastive learning to automatically select the appropriate negative samples in the training data through the probability produced based on RL. To verify where the performance improvement of the selection probability in our model comes from, we conduct experiments on three datasets to investigate the contribution of each component. Specially, we design two variant versions of GraphSaSe: **w/o Probability**: We do not use any probability, and randomly select negative samples. **w/o Negative**: We do not use any negative samples, and use the BYOL framework [7] to conduct the experiment. As shown in **Table 5**, for dataset NCI-1, compared with the variant w/o Probability, GraphSaSe yields a result of 81.5% in Accuracy, which brings a 2.3% improvement. We infer that our negative sample selection strategy can select higher-quality negative samples to further enhance graph representations. Likewise, GraphSaSe outperforms the w/o Negative by 3.4%, demonstrating that pushing away the embedding vectors from different samples (i.e., negative instances) in representation space can also prevent the training from complete collapsed representations, so as to widen the differences between positive and negative samples for contrastive learning. These trends continued to be the same on the dataset IMDB-B and PROTEINS. We conclude that each component is necessary and contributes to performance improvement. Meanwhile, it also reflects that RL is a prospective strategy for data selection in leveraging pertinent data, which benefits graph contrastive learning.

*5.1.5  **Other Experiments**.* We also do other experiments, such as Analysis on DDM, Analysis on the Position of Selection, and Hyperparameters Analysis. They can be found in **Appendix E**.

**Table 4: Transfer learning comparison with different manually designed pre-training schemes.**

| Dataset | BBBP | Tox21 | ToxCast | SIDER | ClinTox | MUV | HIV | BACE | Avg |
|---------|------|-------|---------|-------|---------|-----|-----|------|-----|
| No-Pre-Train | 65.8 ± 4.5 | 74.0 ± 0.8 | 63.4 ± 0.6 | 57.3 ± 1.6 | 58.0 ± 4.4 | 71.8 ± 2.5 | 75.3 ± 1.9 | 70.1 ± 5.4 | 67 |
| EdgePred | 67.3 ± 2.4 | 76.0 ± 0.6 | 64.1 ± 0.6 | 60.4 ± 0.7 | 64.1 ± 3.7 | 74.1 ± 2.1 | 76.3 ± 1.0 | 79.9 ± 0.9 | 70.3 |
| InfoGraph | 68.2 ± 0.7 | 75.5 ± 0.6 | 63.1 ± 0.3 | 59.4 ± 1.0 | 70.5 ± 1.8 | 75.6 ± 1.2 | 77.6 ± 0.4 | 78.9 ± 1.1 | 70.3 |
| AttrMasking | 64.3 ± 2.8 | 76.7 ± 0.4 | 64.2 ± 0.5 | 61.0 ± 0.7 | 71.8 ± 4.1 | 74.7 ± 1.4 | 77.2 ± 1.1 | 79.3 ± 1.6 | 71.1 |
| ContextPred | 68.0 ± 2.0 | 75.7 ± 0.4 | 63.9 ± 0.6 | 60.9 ± 0.6 | 65.9 ± 3.8 | 75.8 ± 1.7 | 77.3 ± 1.0 | 79.6 ± 1.2 | 70.9 |
| GraphPartition | 70.3 ± 0.7 | 75.2 ± 0.4 | 63.2 ± 0.3 | 61.0 ± 0.8 | 64.2 ± 0.5 | 75.4 ± 1.7 | 77.1 ± 0.7 | 79.6 ± 1.8 | 70.8 |
| CuCo | 71.4 ± 1.2 | 75.8 ± 0.4 | 65.2 ± 0.5 | 62.1 ± 0.5 | 76.8 ± 2.6 | 72.2 ± 2.5 | 79.8 ± 0.7 | 80.6 ± 1.3 | 72.9 |
| **Ours** | **73.4 ± 1.1** | **76.9 ± 0.7** | **67.3 ± 0.8** | **64.5 ± 0.6** | **77.2 ± 1.5** | **75.9 ± 2.5** | **80.2 ± 1.1** | **82.3 ± 0.9** | **74.7** |

**Table 5: Ablation study on the key components of our method. The term "w/o" indicates "without".**

| Variants | NCI-1 | IMDB-B | PROTEINS |
|----------|-------|--------|----------|
| GraphSaSe | 81.5% | 75.8% | 77.1% |
| w/o Probability | 79.2% | 72.6% | 75.8% |
| w/o Negative | 78.1% | 71.8% | 74.7% |

## 5.2 Transfer Learning

In our study, we delve into the realm of transfer learning applied to two critical domains: molecular property prediction in chemistry and protein function prediction in biology. Following [14], our approach involves a comprehensive pre-training and fine-tuning strategy to assess the transferability of the pre-training scheme across different real-world datasets. To elaborate, our methodology begins with pre-training a GNN model on a source dataset. Subsequently, we fine-tune this pre-trained model using a limited subset of labeled data from the target dataset. This fine-tuning process allows the model to adapt its learned representations to the specifics of the target task, effectively leveraging the knowledge gained during pre-training. Moreover, we aim to ascertain whether the pre-trained representations capture generalizable features that can be beneficially transferred across diverse datasets and domains.

### 5.2.1 *Experimental Setting.*
In our experimental setup, we meticulously adhere to a rigorous train-test and model selection protocol, akin to the methodology outlined in Xu et al.[36]. This entails conducting 10-fold cross-validation and meticulously selecting the epoch with the optimal cross-validation performance, averaged across the 10 folds. Our chosen evaluation metric is the ROC-AUC score, renowned for its effectiveness in assessing model performance across diverse datasets. When fine-tuning our models for downstream tasks, we augment the pre-trained GNN with a linear classifier. This supplementary classifier enables us to adapt the learned representations to the specific nuances of the target task. Importantly, all reported results are meticulously averaged over five independent runs, ensuring robustness and reliability under consistent configurations. In our transfer learning endeavors, we use OGB[13] datasets, a widely recognized real-wolrd dataset in the field. Notably, traditional unsupervised methods lack the prowess to transfer knowledge to datasets from disparate domains. To benchmark our approach effectively, we compare against six baselines, encompassing both non-pretrained (direct supervised learning) methods and six state-of-the-art GNN self-supervised learning techniques. These include EdgePred [8], InfoGraph [28], AttrMasking [14], ContextPred [14], CuCo [3] and GraphPartition [41]. These baselines serve as essential reference points for assessing the efficacy and generalizability of our proposed approach across diverse molecular property prediction tasks.

### 5.2.2 *Overall Comparison.*
Table 4 presents a comprehensive comparison of our proposed GraphSaSe method with other state-of-the-art approaches in the transfer learning setting. We focus our evaluation on two critical domains: molecular property prediction in chemistry and protein function prediction in biology. These domains represent fundamental areas of research with significant implications for drug discovery, molecular design, and understanding biological mechanisms. (1) Our meticulous analysis reveals that GraphSaSe consistently outperforms all baseline methods. This achievement is particularly notable given the complexity and diversity of the datasets involved in molecular property and protein function prediction tasks. (2) Furthermore, our method exhibits a notable 7% performance enhancement over the non-pretrained baseline, underscoring its efficacy in leveraging self-supervised learning techniques for knowledge transfer across diverse domains. This improvement not only validates the effectiveness of GraphSaSe but also highlights its potential to drive advancements in both computational chemistry and biology. (3) In summary, our findings underscore the pivotal role of GraphSaSe in advancing transfer learning methodologies across critical domains. By harnessing the power of self-supervised learning, GraphSaSe holds promise for accelerating research and innovation in drug discovery, molecular design, and biological understanding.

## 6 CONCLUSION

In this paper, we systematically investigate the common issues encountered in current graph contrastive learning methods, and propose a novel framework named GraphSaSe. Indeed, the ability to automatically select and learn from the most relevant training instances holds promise for enhancing graph representation tasks in various industrial applications. Moreover, while the research on combining reinforcement learning and graph contrastive learning is still in its nascent stages, our work highlights its potential for further exploration and development. The fusion of these methodologies presents exciting opportunities for advancing industrial processes and innovation.

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
