# OpenReview forum: "A Sample-driven Selection Framework: Towards Graph Contrastive Networks with Reinforcement Learning"
_acmmm.org/ACMMM/2024/Conference — MM2024 Poster_

### Official Review · Reviewer_FsqW · 2024-05-07

**Rating:** 3
**Confidence:** 4

**Summary:**

This paper proposes a graph contrastive learning framework named **GraphSaSe** that combines reinforcement learning with graph contrastive learning. The main idea is to use an RL model to adaptively select negative samples during training. The approach consists of two main components: a GCL framework and a selection distribution generator driven by reinforcement learning. Experiments on graph classification and transfer learning tasks show that GraphSaSe outperforms several state-of-the-art graph representation learning methods.

**Reinforcement learning model**:
state: a state represents the current set of selected positive and negative graph instances.
action: determine which samples in the batch are positive, and which are negative.
reward: minimize the divergence for positive pairs while ensuring the selected negative samples are sufficiently dissimilar

**Positive Samples**: Positive samples are generated by applying graph augmentations, just like the traditional GCL paradigm.
**Negative Samples**: Negative samples are selected by the above RL model,

**Strengths:**

1. This novel approach combines reinforcement learning and contrastive learning for adaptive negative sample selection in graph representation learning. This is the first time I see a paper bring RL and GCL together.
2. GraphSaSe uses reinforcement rewards, instead of solely contrastive losses, to minimize the divergence between positive graph samples.
3. The detailed ablation study and hyperparameter analysis validate GraphSaSe's effectiveness.
4. GraphSaSe is an efficient training pipeline, which is only slightly slower than GCN.

**Limitations:**

Below are my main considerations, listed in descending order of importance.
1. The compared baseline methods are weak. I rarely see methods published in 2023 or later. The authors should include more recent state-of-the-art graph pre-training baselines.
2. There's no exploration of alternative reinforcement learning algorithms or reward-shaping strategies. Will the performance gains vary if we change the RL algorithm?
3. Since GraphSaSe still follows the contrastive learning paradigm, the model's performance could influenced by the pre-training and downstream batch size. The RL module may also be sensitive to the number of samples in a batch. The discussion over this hyperparameter is lacking.

**Suitability:**

2

---

### Official Review · Reviewer_mUCZ · 2024-05-24

**Rating:** 5
**Confidence:** 4

**Summary:**

This paper introduces a new graph contrastive learning approach named GraphSaSe, designed to address the problem of negative sampling. The approach comprises two components: Graph Contrastive Learning Framework (GCLF) and Selection Distribution Generator (SDG). By integrating Reinforcement Learning (RL) with graph contrastive learning, the framework dynamically selects appropriate negative samples during training. Extensive experiments across various real-world datasets demonstrate the effectiveness of GraphSaSe.

**Strengths:**

1) The paper studies a key problem in the prevalent graph contrastive learning: positive and negative sampling.

2) The idea of integrating reinforcement learning into graph contrastive learning for dynamic negative sample selection is innovative.

3) The paper is well-written and easy to follow.

**Limitations:**

1) The paper primarily focuses on graph representation learning, with limited discussion on the model's adaptability to other data types or tasks, e.g., images in CV. In my opinion, the proposed approach could extend to tasks and models beyond the current scope. Thus, I suggest that the authors discuss adaptability, any understanding is valuable.

2) The baselines are outdated. Although the proposed approach shows promise, it is crucial to include a comparative analysis with the latest models to validate its effectiveness comprehensively.

3)  The paper lacks a detailed exposition on the selection of hyper-parameters.

**Suitability:**

3

---

### Official Review · Reviewer_GLVB · 2024-06-05

**Rating:** 4
**Confidence:** 4

**Summary:**

The paper presents a novel approach named GraphSaSe, which aims to enhance GCL by integrating reinforcement learning. GraphSaSe consists of two main components: a Graph Contrastive Learning Framework (GCLF) and a Selection Distribution Generator (SDG). The SDG uses reinforcement learning to dynamically select negative samples, optimizing the contrastive learning process. The effectiveness of this approach is validated through comprehensive experiments across diverse real-world datasets, demonstrating its superiority over state-of-the-art methods.

**Strengths:**

1. The integration of reinforcement learning with graph contrastive learning to address the negative sample selection problem is innovative.
2. The paper provides extensive experimental results on multiple datasets, showcasing the robustness and generalizability of the proposed method.
3. The contributions of the paper are well-defined, including the novel GraphSaSe framework and its components, as well as the specific improvements in negative sample selection.

**Limitations:**

1. Could the authors provide a more detailed theoretical justification for the use of reinforcement learning in this context? How does it specifically improve the selection of negative samples?
2. How does the proposed method scale with very large graphs or datasets? Are there any inherent limitations that need to be addressed?
3. Missing important baselines in both the related work part and the experimental part: Bringing Your Own View: Graph Contrastive Learning without Prefabricated Data Augmentations; GPS: Graph Contrastive Learning via Multi-scale Augmented Views from Adversarial Pooling; Graph Self-supervised Learning with Augmentation-aware Contrastive Learning; Self-supervised Graph-level Representation Learning with Adversarial Contrastive Learning.
4. How sensitive is the performance of GraphSaSe to the choice of hyperparameters, especially those related to reinforcement learning?
5. Can the authors provide more insight into how the selection distribution generator operates in various graph contexts? Are there specific types of graphs where it performs better or worse?
6. It is best to provide a code link to enhance the repeatability of the paper.

**Suitability:**

2

---

### Official Review · Reviewer_oc8v · 2024-06-06

**Rating:** 4
**Confidence:** 2

**Summary:**

This paper mainly focuses on improving negative sampling in the graph contrastive learning framework. Specifically, it adopts reinforcement learning to select the most appropriate negative samples, i.e., those furthest away from the positive pairs. It regards the differences observed between positive graph representations as the reward function. Then a negative sample selection distribution generator is devised, enabling adaptive selection of negative samples to effectively guide the representation learning process.

**Strengths:**

1. The paper is the first to utilize RL to select the appropriate negative samples in graph contrastive learning.
2. The experiments are very extensive, proving the effectiveness of the method.

**Limitations:**

1. Assumption about selecting distant negative samples:
The assumption that we should select negative samples that are far away from the positive samples, which requires further theoretical or empirical support. Previous work [1] has suggested that "hard" negative samples (similar to positives but with different labels) can be more beneficial for contrastive learning.
The authors need to clarify the validity of their assumption and position it in the context of related findings. They should provide a stronger justification for why distant negative samples are preferable in their framework, compared to other negative sampling strategies.
2. Clarity of Figure 1:
The current Figure 1 is not informative enough. It is unclear what the authors want to emphasize with this figure.
If the goal is to show that the proposed method can avoid selecting nodes with the same class as negatives, this does not seem well-aligned with the claimed motivation of minimizing the similarity between positive and negative batches.
The authors should either improve the figure to better illustrate the key ideas or consider removing it if it does not significantly contribute to the paper's narrative.
3.Time complexity analysis:
There is a good observation that the proposed method does not seem to have high time complexity, as it can even outperform GraphCL in some cases (as shown in Table 3).
The authors should provide more analysis and description in the experiments section to explain the factors that contribute to the reduced time complexity. A deeper exploration of the time complexity analysis and comparison to related methods would strengthen the technical contributions of the paper.
4. Model optimization:
It's not clear whether the distribution generation policy and graph representation learning are jointly optimized or optimized in an alternating manner.
The authors should clarify the details of the optimization procedure in the methodology section.
5. Paper writing improvements:
the introduction section seems unnecessarily verbose, especially the long list of potential application cases.
The authors should streamline this section and focus on concisely explaining how improved negative sampling techniques can benefit graph representation learning and downstream tasks, without going into excessive detail about specific applications.
Regarding the figures, the authors should ensure that the font size and overall presentation are clear and readable. Adjusting the font size in Figure 1, would be a good improvement.
[1] Generating Counterfactual Hard Negative Samples for Graph Contrastive Learning

**Suitability:**

2

---

### Official Review · Reviewer_Bu9q · 2024-06-07

**Rating:** 5
**Confidence:** 2

**Summary:**

The authors developed GraphSaSe, a novel Graph Contrastive Learning (GCL) method that addresses the limitations of traditional semi-supervised learning by using reinforcement learning to adaptively select negative samples. It features two key components: a Graph Contrastive Learning Framework (GCLF) and a Selection Distribution Generator (SDG). The method translates the divergence between positive graph representations into a reward mechanism to guide negative sample selection dynamically, enhancing graph representation learning. Extensive experiments validate its effectiveness, showing significant improvements over existing methods.

**Strengths:**

1. Adaptive Negative Sample Selection: GraphSaSe uses reinforcement learning to adaptively select negative samples, overcoming the limitations of traditional methods that rely on random sampling, thus improving model performance.
2. Innovative Approach: It introduces reinforcement learning for negative sample selection in graph contrastive learning, providing a novel approach to this challenge.

**Limitations:**

1. The reinforcement learning component may require extensive parameter tuning to ensure the effectiveness of the selection distribution generator, necessitating additional experiments and time.
2. The authors may consider providing an analysis of the class distribution balance in the existing datasets (optional) and offer insights and discussions on the application of this method to imbalanced data distributions.

**Suitability:**

2

---

### Meta-Review · Area_Chair_J9Jr · 2024-06-26

**Recommendation:** Accept (Poster)
**Confidence:** 4

**Metareview:**

This paper received positive scores (ba,ba,ba,wa,wa) from all of reviewers after rebuttal. This paper introduces reinforcement learning for graph contrastive learning. Reviewers believe that the method is innovative and the experiments are extensive. There are also some concerns from reviewers such as weak baselines and lack of discussion in experiments. Authors should solve these issues in their final version.